# Morphological and Genetic Mechanisms Underlying Awn Development in Monocotyledonous Grasses

**DOI:** 10.3390/genes10080573

**Published:** 2019-07-30

**Authors:** Fabrice Ntakirutimana, Wengang Xie

**Affiliations:** State Key Laboratory of Grassland Agro-Ecosystems, Key Laboratory of Grassland Livestock Industry Innovation, Ministry of Agriculture, College of Pastoral Agriculture Science and Technology, Lanzhou University, Lanzhou 730020, China

**Keywords:** awns, morphology, histology, genetic basis, awn development, grasses

## Abstract

The identification of biological mechanisms underlying the development of complex quantitative traits, including those that contribute to plant architecture, yield and quality potential, and seed dispersal, is a major focus in the evolutionary biology and plant breeding. The awn, a bristle-like extension from the lemma in the floret, is one of the distinct morphological and physiological traits in grass species. Awns are taught as an evolutionary trait assisting seed dispersal and germination and increasing photosynthesis. Awn development seems to be complex process, involving dramatic phenotypic and molecular changes. Although recent advances investigated the underlying morphological and molecular genetic factors of awn development, there is little agreement about how these factors interact during awn formation and how this interaction affects variation of awn morphology. Consequently, the developmental sequence of the awn is not yet well understood. Here, we review awn morphological and histological features, awn development pathways, and molecular processes of awn development. We argue that morphological and molecular genetic mechanisms of awn development previously studied in major cereal crops, such as barley, wheat, and rice, offered intriguing insights helping to characterize this process in a comparative approach. Applying such an approach will aid to deeply understand factors involved in awn development in grass species.

## 1. Introduction

In monocotyledonous cereal crops, such as rice (*Oryza sativa* L.), wheat (*Triticum aestivum* L.), barley (*Hordeum vulgare* L.), and rye (*Secale cereale* L.), as well as several forage grasses, the inflorescence is characterized by bristle-like extensions, called awns, adhered at the tip end of the lemmas in the florets [1]. Awn formation and development rely on lemma primordium, which produces awn meristem as the lemma apex continues to elongate [2]. The sequence of awn development events is a complex process, involving morphological features, such as inflorescence density and glume length to control awn length [3], growing conditions [4], and genetic mechanisms [5]. Awns are beneficial in wild species because they aid in seed dispersal, as the barbed awns adhere the seed to animal fur [6]. They repel seed-eating birds and animals. Long awns are also easily controlled by external dispersal factors such as wind, and their movement may propel the seed into the ground, which enables self-planting [7]. Awns are also effective for light interception and CO_2_ uptake because photosynthates travel easily from awns to the growing kernels [8]. Evidence in barley [1,5] and in wheat [8,9] revealed that awns are coupled with grain yield increase, particularly in drier and warmer environments. Despite the undisputable importance of awns especially in nature, long and barbed awns deter manual harvesting and interfere with seed-processing activities, that is, malting and milling [10]. In forage barley cultivars, long and barbed awns increased bulky fibers, hindering forage quality and palatability [11].

In transverse sections, the awns of barley, wheat, oat, and rye are triangular-shaped with three vascular bundles and two strands of chlorenchyma tissues, and they may contribute to photosynthesis. Awns of rice, on the contrary, have a round shape in transverse section with single vascular bundles and a lack of chlorenchyma tissues, implying their minor contribution to photosynthesis [1]. Supporting this observation, awn removal experiments revealed a minor impact on grain yield and, consequently, rice species with shorter or nonexistent awns were preferred by early cultivators to enhance harvesting and postharvest processing. This is in contrast with some other domesticated cereals, such as barley and wheat, where the removal of awns led to yield loss. Thus, most cultivated barleys and wheats possess long awns [12].

Previous genetic studies identified several awn-related quantitative trait loci (QTLs) in several major cereals, and several genes were recently cloned in some. In this respect, the genetic basis of awn development was deeply investigated in rice, and allowed researchers to identify many QTLs associated with awn growth on almost all chromosomes [13,14]; furthermore, some of their causal genes were dissected molecularly [12,13,14,15,16,17]. In barley, functional advancements regarding genetic mechanisms of awn development emerged recently, with the cloning and characterization of genes such as *Lks1* [18] and *Lks2* [1]. Although genes causing awn development in wheat are not yet cloned, scientists identified the main inhibitors of awn development in some wheat cultivars [19].

Despite the undisputable role of awns for inflorescence architecture and their putative impact for yield determination, details regarding how the awn is formed and how its morphology is variable among species remain elusive, and a clear model explaining the awn developmental process is needed. As such, it demands solid and basic skills of the major plant developmental processes. Our aims in this review are (1) to discuss the phenotypic evolution of awns by illustrating morphological structures of the awn and physiological changes during awn development, and (2) to assess the genetic basis associated with awn development and variation in major cereal crops. Understanding awn development is essential for agriculture, with respect to improving yield and quality traits of grass species.

## 2. Morphological and Histological Characteristics of the Awn

### 2.1. Morphological Features of the Grass Inflorescence

The grass inflorescence is a unique, complex structure, different from those of most other plants. It comprises one to several structural units called florets, packed together in a spikelet [20]. The spikelet is the basic unit of the grass floral structure, usually comprising glumes (either empty glumes or rudimentary) and one to several florets. The floret, the spikelet’s individual unit, consists of a lemma (which extends in the functional awn in some grasses), a palea, and reproductive organs, including the stamens, pistils, and lodicules [21]. The arrangement of spikelets and the number of florets are key determinants of inflorescence type in grass species. Based on the arrangement of spikelets, the three most well-known kinds of grass inflorescence are spike, raceme, and panicle. A spike is an unbranched inflorescence in which spikelets are attached directly to the main axis, as in wheat, barley, and rye. A raceme is an unbranched inflorescence in which spikelets are attached to the main stem by pedicels, as in *Festuca*, some *Danthonia*, and *Avena*. A panicle is a branched inflorescence with spikelets attached to the side branches rather than the main stem with or without pedicels, such as what is found in rice and oats [22]. The inflorescence meristem may be classified based on floral meristem determinacy. The inflorescence is determinate, when the main axis always ends in an apical spikelet, as in wheat, rye, and oats. The inflorescence is indeterminate, when the main axis never becomes terminated by a spikelet and continues to initiate branches and a number of fertile spikelets, as in barley and maize [2,23]. 

Spikelets are modified units defining the inflorescence structure of grass species. Based on the presence or absence of pedicel, a small stalk or stem connecting the flower to the main axis, spikelets may be sessile or pedicelled. Sessile spikelets are connected directly to the axis, whereas pedicelled spikelets are attached to the axis by the stalk [22]. Scientists also classified spikelets referring to the position of the spikelets in the main inflorescence. Basal spikelets are located in the bottom, central spikelets in the center, and apical spikelets on the top of the inflorescence [23]. Variation of the grass floral morphology contributes greatly to the diversity of grass spikelets. For instance, the presence of fertile florets (those possessing a caryopsis) and sterile florets (empty glumes) raised arguments in the classification of spikelets. A fertile spikelet contains one to several fertile florets, whereas a sterile spikelet contains only sterile florets [2,23]. Furthermore, the grass inflorescence is usually composed of both male and female structures. In some cases, the spikelet may possess imperfect flowers with only male or female structures. For instance, the staminate spikelet only contains stamens, whereas the pistillate spikelet only possesses pistils [24]. Spikelets can also be classified based on the presence or absence of awns. The awned spikelet possesses at least one but sometimes several awns, while the awnless spikelet possesses no awn in any form [25,26,27].

### 2.2. Awn Anatomy and Histology

Awns are characteristic features of grass inflorescence [7]. While awns’ morphological features vary among grass species, their description varies among sources. Most cultivated cereals (for example, wheat, barley, oat, and rye) are awned and possess a single awn per floret, but some cereal grasses, such as maize and sorghum, are awnless [19,25]. In rice, most wild rice species are known as long-awned, whereas cultivated cultivars are awnless or remarkably short-awned [6]. In addition, some wild grasses may have two to several awns per floret, as in grasses from the *Aristida* genus [28,29]. Thus, awn length is one of the main distinguishing features in grasses, and domestication process via awn length led to dramatic phenotypic variations through evolutionary processes [6,15]. There are several arguments in deciding whether the awn is long or short. In barley, awns that are more than twice the length of the spike are classified as long, whereas awns that are less than twice the length of the spike are classified as short [30]. In wheat, there is no rule for awns to be long or short, and these categories are estimated via observations [31].

Straightness is also another distinguishing feature of grass awns. Awns that are extremely bent or flex from the middle to the tip end are known as curved, whereas awns that are straight or slightly curved are classified as straight. Also, long awns are usually curved, and short awns are straight [19,32]. Furthermore, the presence of barbs on the outer layer of the awn is also a distinguishing feature in grasses. Rough (or barbed) awns are entirely covered with barbs, whereas smooth (or barbless) awns have no barbs on the outer surface [11,15,32].

In different species, awns are distinct in cell type, structure, and number. Long awns have more elongated cells in the top and bottom and less in the middle, whereas short awns contain small cells, and most of the cells occupy the middle part [1]. Except for rye and its relatives lacking chlorenchyma cells, many species such as wheat and barley have parenchyma cells on both awn sides, and parenchyma cells occupy a big part of the awn transverse section (Figure 1). Longitudinal sections of the awn show clear branched and elongated chlorenchyma cells with two or more roundish proteinaceous bodies [33]. Moreover, several grasses contain stomata in their awn’s external tissues, but some cereals such as wheat and rice contain cuticles in the awn external layers. Long awns also possess more phytoliths (silica depositions) and adaxial and abaxial terminals in comparison to short awns [34,35].

Awn cells are responsible for several functions assisting in the elongation of the awn, as well as the development of the seed grain. Parenchyma cells and green cells (chlorophyll-containing cells) synthesize photosynthate; vascular bundles work together with parenchyma cells to convey assimilates to the other cells; abaxial and adaxial terminals and stomata regulate water, temperature, and metabolic chemicals through transpiration; proteinaceous bodies mainly store proteins and minerals; phytoliths maintain the rigidity and persistence of the awn [36]. It is noteworthy that improvements in the performance of cereal crops and forage grasses require an in depth understanding of plant morphological features including those of seed awns.

## 3. Awn Development Pathways

Plant development analyses revealed that the reproductive phase is simpler in eudicots, such as *Arabidopsis thaliana* (L.) Heynh, where the inflorescence meristem (IM) directly generates the floral meristem (FM) unlike in monocots, such as grasses, where the process can undergo various intermediate stages that may last several weeks [2]. Although the reproductive process is more complex in grasses, the vegetative phase is nearly the same. The plant developmental process in grasses is governed by the shoot apical meristem (SAM) and the root apical meristem (RAM), meristems formed during embryogenesis. The SAM controls the development of leaves, stems, and flowers, whereas the RAM controls the rooting system. During vegetative growth, the SAM generates leaf primordia; the leaf primordia produce the axillary meristems (AMs), and these meristems generate the secondary shoots or tillers. After the vegetative growth, the SAM shifts its focus to the inflorescence meristem (IM) and, thus, the reproductive phase is initiated [2,37]. The sequence of reproductive events varies in grass species, and it involves various intermediate meristems. To better understand the process of awn development, below, we compare the developmental fate of rice and wheat, two major grasses, exhibiting different developmental processes.

In rice, florets are developed by spikelet meristems (SPMs) which are initiated by the branch meristems (BMs), that is, either by primary branch meristems (PBMs) or secondary branch meristems (SBMs). The IM generates PBMs, which then develop SBMs. The branch meristems (BMs) generate the SPMs including lateral spikelet meristems and terminal spikelet meristems, SPMs generate FMs, and FMs initiate spikelets. The empty glumes begin spikelet development, followed by the lemmas, paleas, and lodicules; then, fertile florets produce the stamens and pistils [21]. 

The development of awns in rice is one of the intermediate floral developmental phases. Figure 2 is a generalized diagram of awn development in rice. It also demonstrates several parts of the spikelet. The initiation of the awn can be defined as the morphological change of the lemma apex as floral organs continue to grow. The awn primordium is initiated after the lemma primordium [16]. This stage is followed by the appearance of a pair of ridges, in which the upper ridge generates the panicle and its parts. As the process proceeds, the lemma continues to grow, and the awn propagates from the terminal part composing the vascular bundle of the lemma [21]. Although this process clearly shows that awns protrude from the lemma, problems regarding factors causing awn suppression in some rice species persist. Luo et al. [12] used both awned and awnless rice cultivars to illustrate awn development. They found that both awned and awnless cultivars exhibit similar growth before the lemma primordium. After the lemma primordium, the lemma apex of the awned cultivar elongates, and the awn primordium differentiates and extends. In contrast, the lemma apex of the awnless cultivar stops growing and forms a round tip. Thus, awn development in rice is divided into two main stages: awn formation, which begins after lemma primordia and ends at stamen primordia; and awn elongation, which begins during stamen primordia and ends during early maturity, the dough stage [16].

In wheat, the floral developmental process is more complex. Unlike rice having a single floret per spikelet, wheat spikelets contain three to six florets [38]. This variation in spikelet patterning may affect the developmental processes. After the shift from the vegetative to reproductive phase, the IM initiates the spike meristem (SM). This phase is indicated by the appearance of a pair of ridges. The upper ridge represents the SPM, and the lower ridge forms the leaf primordium. Both ridges continue to differentiate until the upper covers and suppresses the lower. As the process continues, the SM initiates spikelet meristems (SPMs) starting in the middle toward the base and the top of the floral body. After the initiation of FMs by SPMs, two pairs of glumes differentiate. The first pair appears on both sides of the SPM as transverse ridges, and the latter appears above the first pair. The first pair becomes empty glumes, and the second differentiates into the lemmas of the first and second florets. As the development proceeds, the FM elongates upwardly (from the base to the apex) until the initiation of the lemma of the sixth floret. The first two florets generate anthers, followed by the paleas and then the pistils. The third floret generates its parts, followed by the fourth, fifth, and sixth floret, respectively. In the awned species, lemmas continue to elongate and develop the awns. The development of awns within the spikelet follows the fate of their respective florets, that is, starting in the first floret to the sixth [23,37].

The study by Bonnett [38] unveiled that all parts of the wheat floral structure including awns become visible before the differentiation of the last floral organ, the peduncle. Despite this, the initiation time of the awn primordium remains poorly understood. The study by Ponzi and Pizzolongo [39] in common wheat cv. Ofando found that SPMs and subtending leaf primordia expand and initiate FMs, the process probably followed by the initiation of awn primordia. The authors revealed that the initiation of awn primordia at this stage was indicated by the presence of the positive starch granules (PSGs) at the base of the developing spikelet. As the development proceeded, FMs initiated floral primordia at the axil of the lemma, while awns expanded rapidly at the top of the lemmas. The authors also found that awns and anthers might be initiated at the same time because PSGs were also found in the anthers, but were absent in other floral parts. This finding is consistent with the study by Bonnett [40] in barley and Bremer-Reinders [41] in rye. In both species, the basal of the lemma differentiates rapidly and initiates awn primordia during the initiation of anther primordia. After the initiation of floral parts, the awn expands more rapidly than the lemma and palea to attain a considerable length. Furthermore, the elasticity in the duration from awn primordium to the time when the awn is apparent raised several arguments. A study suggests that this duration is controlled by genetic mechanisms and growing conditions such as vernalization, temperature, and photoperiod [42].

Awn development varies considerably in grass species despite possible little similarities. In the *Danthonieae* tribe, the lemma is divided at the apex into three lobes, in which the middle one extends into a geniculate awn. In the *Aveneae* tribe, the awn arises on the back of the lemma with oneor two lobes [43]. This variation is regulated by several genetic switches that may initiate the awn primordia, inhibit awn differentiation in awnless species, control the duration of the process and, thus, control the interaction of awns and several signaling centers [12,16]. It is, thus, noteworthy that unveiling the molecular underpinnings of awn development remains important.

## 4. Genetic Basis of Awn Development in Grasses

As the awn exhibits complex morphology and development [14,44], identifying its growth regulators was a challenge for decades [1]. To address this, focus should be put on the genetic mechanisms given their considerable influence in the development of awns [6,12,37]. This section reviews strategies that were applied so far to identify quantitative trait loci (QTLs) and causal genes for awn development. It also discusses how these efforts could provide novel insights into understanding awn development and how these insights may trigger ongoing and new research initiatives.

### 4.1. Awn Development and Recent Genetic and Genomic Advances

The artificial selection by ancient farmers was considered the root of the reduced awn length in cultivated grasses [5]. However, questions on the variability of awn features among and within grass species persist. Moreover, the interaction of awns and other domestication traits such as panicle architecture, grain size, grain weight, and seed shattering obscure the genetic basis of awn development and, correspondingly, may have far-reaching economic implication in several grass species [45,46]. To resolve this puzzle, scientists adopted several approaches to reveal the genetic basis of awn development.

QTL mapping (traditional linkage mapping) was an important tool in chasing awn development regulators for many years [47]. QTL mapping depends on molecular markers such as restriction fragment length polymorphisms (RFLP), amplified fragment length polymorphism (AFLP), sequence-related amplified polymorphism (SRAP), simple sequence repeat (SSR) or single-nucleotide polymorphism (SNP), and mapping populations, including F_1_, F_2_, backcross (BC), recombinant inbred lines (RIL), doubled haploid (DH), and multiparent advanced generation intercross (MAGIC) lines [48]. Several studies adopted QTL mapping and identified a huge number of QTLs that control awn development in major grasses such as rice, wheat, and barley [49,50]. QTL mapping enabled scientists to develop recombinant inbred lines (RILS), which were used to illustrate awn development in grasses (for example, in rice) and disclose the interaction of awns and yield traits [51]. However, unlike rice species, compatible with this tool, defining a few megabases and covering thousands of genes was a hindrance in many species, such as polyploidy wheat and barley, having huge and complex genomes [48]. To resolve this issue, scientists envisaged several other approaches, such as fine mapping and QTL cloning. These approaches displayed several QTLs underlying awn development in rice [49,51], barley [18], wheat [19], and sorghum [52] despite being a time-consuming and expensive approach. They also assisted to dissect several causal genes for awn development in some major cereals such as rice [12] and barley [1].

The interactions among awn development QTLs and grain yield QTLs and the environment were a drawback for breeding strategies in grasses. Therefore, QTL cartographer (that is, a software suite which uses molecular markers to map quantitative traits) and QTL networking (that is, bioinformatics-based software which analyzes QTL data of large genotypes and phenotypes collected from many groups of related individuals) are alternatives to resolving this issue [53]. Recently, research by Masoud et al. [50] applied these approaches and intriguingly found six additive awn length QTLs using QTL cartographer and three additive and two epistatic QTLs using QTL networking. 

Genome-wide association analysis (GWAS) offers a new approach to gene discovery unbiased with regard to presumed functions or locations of the causal variant. It was a powerful tool in displaying QTLs and genes underlying awn development. This approach applies linkage disequilibrium (LD) to unveil polymorphisms associated with a trait of interest [48]. Using this approach, Magwa et al. [46] identified several QTLs linked to awn length and one QTL linked to *AN1* (the main gene controlling awn development in rice). Interestingly, they also identified an association SNP in a region near the seed shattering gene (*qSH1*), indicating that awn development and seed shattering may exhibit molecular interactions. Finally, these approaches enabled researchers to identify several genes and QTLs underlying seed awning especially in major cereals. Thus, additional advances are required to further explain awn development in several species, particularly forage grasses, and unveil the interaction between awns and other seed traits. 

### 4.2. Molecular Mechanisms Underlying Awn Development in Grasses

#### 4.2.1. Mechanisms of Awn Development Causal Genes

Awn development is a polygenic trait that is regulated by many genes, with several genes identified in rice, along with their role in the initiation and growth of the awn (Figure 3; Table 1). Although awns have a negligible effect on rice performance [1], this species received considerable interest in displaying the genetic basis of awn development due to its small genome size and its relationship with several other cereal crops. Additionally, rice species exhibit variation in awn features. For example, wild rice species such as *Oryza rufipogon* Griff. and *Oryza barthii* A. Chev. are long-awned. Most cultivated rice species such as *O. sativa* cv. Indica and *Oryza glaberrima* Steud. are awnless, whereas *O. sativa* cv. Japonica is short-awned [6]. This variability reflects that awn development in rice is controlled by drastic genetic changes. Matsushita et al. [13,14] used the introgression lines generated from wild rice (*Oryza meridionalis* Ng.) and cultivated rice (*O. sativa*) and uncovered five genes (*An6*, *An7*, *An8*, *An9*, and *An10*) for awn development in *O. meridionalis*. The authors discovered that *An6*, *An7*, and *An8* generated longer awns than other genes and *An10* generated a short-awn phenotype.

Several years later, Luo et al. [12] used genetic mapping in *O. rufipogon* and *O. sativa* to identify further mechanisms underlying awn development. They found some major genes related to cytokinin regulated awn development. *AN1*, the major domestication gene located on chromosome 4, encodes a basic helix-loop-helix (bHLH) transcription factor linked to awn development in wild rice (*O. rufipogon*). The authors also used in situ hybridization to unveil the cause of awnless habits in cultivated rice (*O. sativa*), and they surprisingly found the *an1* allele in *O. sativa*, which reduces cytokinin in awn primordia and, thus, inhibits awn differentiation. This indicates that the absence of awns in cultivated rice may be caused by the mutation in *AN1* during awn elongation. The authors also observed that the expression of *AN1* in wild rice ceased during the late flowering stage, suggesting that the further elongation of awns after flowering might be induced by other genes. In this respect, Gu et al. [16] revealed that the complete action of *AN1* in wild rice relies on the mutual effect with other genes. Intriguingly, they cloned *AN2* (*AWN2*), a major gene that encodes a *Lonely Guy Protein 6* (*OsLOGL6*), which catalyzes the final step of cytokinin synthesis and, thus, promotes awn elongation in *O. rufipogon.* The authors also stated that the mutual effect of *AN1* and *AN2* could remarkably promote awn development in *O. rufipogon*, implying that *AN1* controls the initiation and formation of awns, whereas *AN2* regulates the elongation of awns. The authors also suggested the awnless phenotype in some rice species may be caused by the suppression or mutation in *AN1*, while the short-awned phenotype may be caused by the mutation in *AN2*. In conclusion, cytokinin is one of the important plant hormones promoting cell division and differentiation during plant organ development. Rice *an1* mutant showed a reduced cytokinin level in awn primordia, while *AN2* encodes a cytokinin synthesis protein *OsLOGL6*, which further confirmed the role of cytokinin in awn development at the molecular level.

In barley, awn development is mainly controlled by *Lks1* and *Lks2* genes located on chromosome arms 2HL and 7HL, respectively. *Lks1* (*Awnless1*) is a dominant gene, which causes the awnless phenotype in barley [59]. *Lks2* (*LONG AWN2*) encodes the SHORT INTERNODES (SHI) family transcription factor that promotes awn elongation [1]. By contrast, its mutant gene, *SHORT AWN2* (*lks2*), reduces number of longitudinal cells in the awn and, thus, shortens awn length up to 50% [5]. 

In wild wheat (*Aegilops tauschii* Coss.), Nishijima et al. [18] identified the awnless locus (*Antr*) on chromosome arm 5D, which inhibits awn formation and, thus, affects the awnless phenotype in wild wheat. The identification of this novel gene provided intriguing insights into the genetic basis of awn development in wheat species. However, further investigation is needed to unveil whether this gene is involved in the development of awns in cultivated wheat.

#### 4.2.2. Functions of Floral Meristem Genes 

The shift from vegetative to reproductive development in grass species is under strict genetic control, in which each pathway involves genetic factors that affect the architecture of secondary inflorescence meristems. Genetic advances reported ample evidence that gene mutations may obstruct the growth of floral meristems and thus transform floral parts into functional awn-like structures. In rice, the growth of shoot apical meristem is regulated by many genes, including *SHOOTLESS2* (*SHL2*), *SHOOT ORGANIZATION1 (SHO1*), and *SHOOT ORGANZATION2* (*SHO2*). Genetic experiments revealed that these genes have strong mutant alleles (*shl2*, *sho1*, and *sho2*) that encode proteins acting in trans-acting small interfering RNA, acting in the verification and silencing of some plant genes and, thus, inhibiting the growth of apical shoot meristem. This process may cause plant abnormalities, including the transformation of the lemma and floral meristems into awn-like structures [63,64]. *TOB1* (*TONGARI-BOUSHI1*), *DL* (*DLOPPING LEAF*), and several other members of the *YABBY* gene family encode transcription factors that are involved in the upholding of plant meristems and development of floral organs in rice. Interestingly, *DL* interacts with *OsETT2* of the *ARF* gene family to initiate cell divisions in the awn primodium and, thus, regulate awn length; *tongari-boushi1* (*tob1*) mutants possess similar functions, but also control the formation and elongation of lemma and palea [65,66].

Evidence also shows that the overexpression of genes that are involved in the expansion of floral meristems can control awn development. In wheat, for example, *Wknox1a*, *Wknox1b*, and *Wknox1d* genes are located on the 4AS, 4BS, and 4DS chromosome arms, respectively. In some wheat cultivars, such as Chinese spring, the overexpression of these genes in the floral meristem may actuate awn formation [67]. This supports the idea that, although the causal genes for awn development arenot yet cloned in cultivated wheats, this process is controlled by several interacting genes. The current data suggest that variation in awn length and hood phenotype in wheat is a result of alterations in genes underlying floral meristems [19]. 

#### 4.2.3. The Role of Interacting Genetic Factors

Several molecular experiments showed that awn development is controlled by multiple interacting genetic factors. In barley, for instance, *LONG AWN2* (*Lks2*) and *HvKNOX3* genes and *suK* alleles (*suKD*, *suKB*, *suKC*, *suKE*, and *suKF*) exhibit molecular interactions, resulting in the variation of awn length. The *HvKNOX3* and *suK* alleles possess a dominant mutant K, which either shortens awn length or inhibits awn formation [62,68]. This dominant mutant K may also divert the initiation and growth of awn meristem and forms a hooded (flower-like) phenotype [60]. Other studies observed that the mutation in *suK* alleles and *Lks2* gene may inhibit the expression of *HvKNOX3* and, thus, promote awn development [19,21,62]. Furthermore, recombinant inbred lines (RIL) from two barley accessions, Azumamugi and Kanto Nakate Gold, displayed a QTL for awn length near the *vrs1* locus, which is known to be involved in regulating spike development and plant statute traits. Molecular analyses revealed that interactions between these genetic factors have pleiotropic effects that result in reduced awn length [67]. Genomic studies in barley also detected molecular interactions between *vrs1* and *Awnless1* (*Lks1*) locus and mutants for reduced awn length, which caused awnless and short-awned phenotypes, respectively [67]. Analysis of the gene controlling the spike density in barley, *dsp1*, revealed that interaction of this gene with the *short awn 2* (*lks2*) gene underlays short and thin awns in Aizu Hadaka 3, a six-rowed barley variety [69]. These observations support the view that awn length in barley is controlled by a wide variety of molecular interactions. They also leave open interesting questions on the molecular interactions among *vrs1*, *lks2*, *dsp1*, and *suK* alleles. Furthermore, a recent study detected *Vrs2*, the gene responsible for the floral organ patterning in barley, in several floral parts including the lemma [20]. Identifying whether this gene is expressed in the awn remains a goal for future studies.

#### 4.2.4. Regulation of Awn Formation and Length by Dominant Inhibitors

In wheat, awn development is strictly controlled by three dominant inhibitors: *Hooded* (*Hd*), *Tipped1* (*B1*), and *Tipped2* (*B2*). Molecular advances detected *Hd* on chromosome arm 4AS, *B1* on chromosome arm 5AL, and *B2* on chromosome arm 6BL [19]. Full-awned phenotypes carry three homozygous recessive (*hd*, *b1*, and *b2*) alleles. Awnless phenotypes possess two dominant alleles and one recessive allele, that is, either *HdHdb1b1B2B2* as in the Chinese spring cultivar or *hdhdB1B1B2B2* as in the Federation cultivar [19,70]. Furthermore, Yoshioka et al. [19] used RILs derived from short-awned common wheat cultivars (Chinese Spring and Mironovskaya 808) to investigate the further impact of these inhibitors on the variability of awn traits. Intriguingly, they found that wheat RILs that possess *Hd* developed very short awns that are bent at the base. Wheat RILs with *B1* generated short awns at the base and middle of the spike, but awn length rose at the top of the spike reaching 1cm. RILs with *B2* produced short, curved awns that were not bent at the base as in *Hd*. Despite the undisputable role of these inhibitors during awn development in wheat, none were characterized molecularly. A study attempted to associate *Hd* with the *Wknox1a* gene. Also, *Hd* and *B2* were found near the *DL* and *RAE2* genes, respectively; however, neither of these genes is directly coupled with awn development in wheat [19]. Thus, future investigations should continue to improve our limited knowledge on molecular genetic factors coupled with awn development in wheat. 

## 5. Conclusions and Future Perspectives

Awn development was described morphologically and genetically, especially in major cereal crops like rice, wheat and barley. These advancements not only offer an intriguing paradigm for studying the evolution and domestication of grasses, but also have significant implications for future breeding programs. From a breeding standpoint, identifying the interactions between seed characters and agronomic traits, including those determining yield and quality potential, remains an essential goal to realizing the potential of agricultural research and innovation. In this respect, combining morphological- and molecular-based approaches will yield comparative data that will help to unveil the pleiotropic impacts of awns in grass species. All these essential perspectives should enable us to unravel the link between awn morphology and several genetic factors in major cereal crops, but the question regarding how we can achieve the same level of understanding in forage grasses remains. In the future, these important genes for awn development identified in major crops, such as *AN1*, *AN2*, *GAD1*, *LABA1*, *Lks1*, and *Lks2*, can effectively be used as a source of candidate genes for other grass species, and they can also facilitate comparative genome research between crops and forage grasses. The application of the information presented here should considerably aid in this quest. 

## Figures and Tables

**Figure 1 genes-10-00573-f001:**
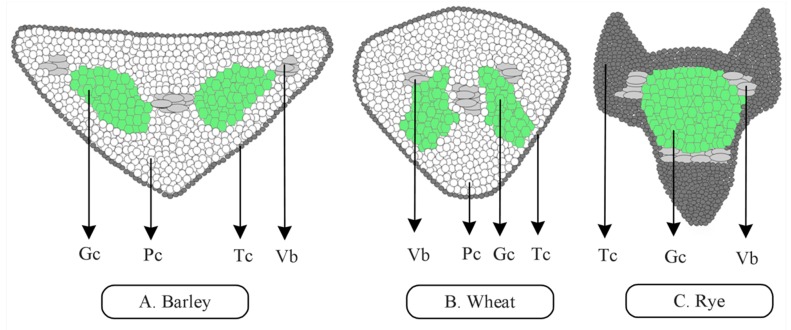
Transverse section shows several cell types. Except for rye lacking parenchyma cells, wheat and barley are similar in most of the cell types despite differences in awn structure. (**A**,**B**) In transverse sections, barley and wheat awns are triangular and show a high number of parenchyma cells compared to other cell types. (**C**) Rye awns are triangular in transverse section with a high number of green cells and thick-walled cells compared to other cells. Gc, green cell; Pc, parenchyma cell; Tc, thick-walled cell; Vb, vascular bundle.

**Figure 2 genes-10-00573-f002:**
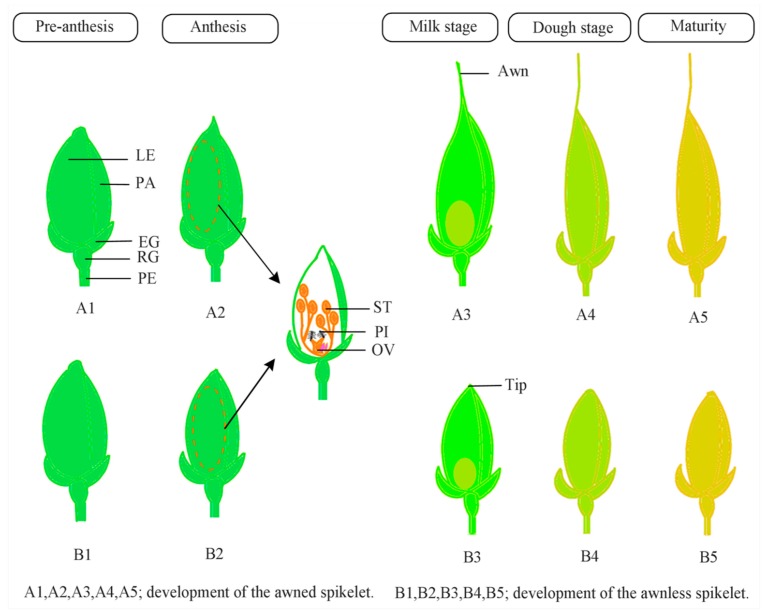
A generalized diagram of awn development in rice. From heading to a few weeks before anthesis, awned and awnless spikelets show similar growth. During the anthesis, reproductive organs differentiate and awned spikelets show higher growth than awnless spikelets. During the milk stage, the grain starts to develop, and the awn elongates in the awned spikelet, whereas the awnless spikelet forms a round tip. During the dough and maturity stages, the awn photosynthetic activity drops, the spikelet decreases in volume, and the awned plant provides a longer and larger seed than the awnless spikelet. EG, empty glume; LE, lemma; OV, ovary; PA, palea; PE, pedicel; PI, pistil; RG, rudimentary glume; ST, stigma.

**Figure 3 genes-10-00573-f003:**
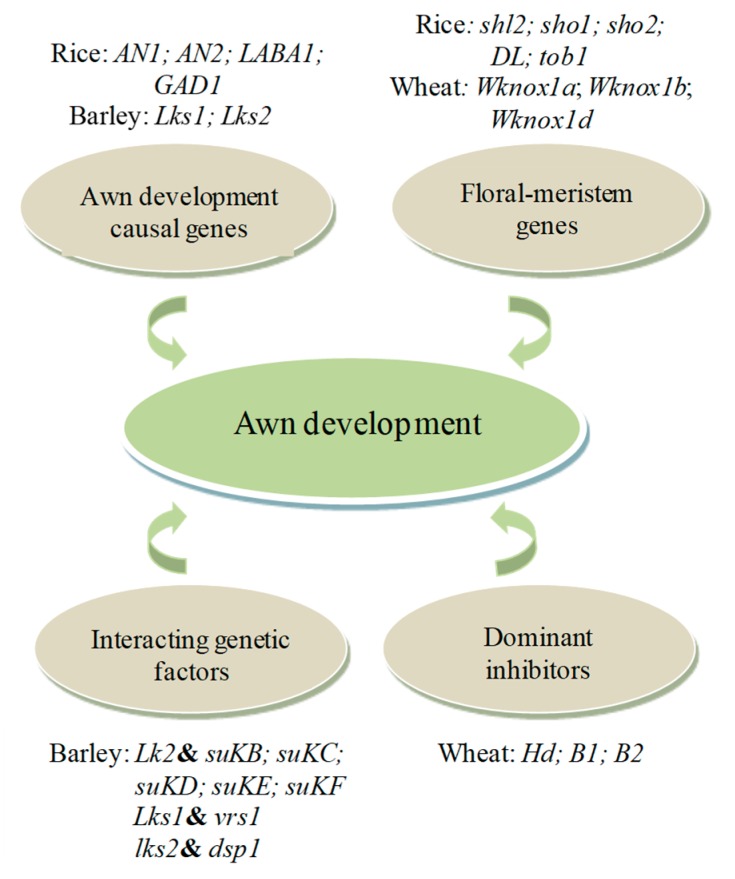
An overview of genetic factors underlying awn development in major cereal crops. The sketch shows major loci that control the initiation and elongation of the awn in rice, wheat, and barley, along with several genes that rely on mutual effects with other genes to promote awn growth. It also shows loci that reduce awn length or inhibit awn development. Both gene expression and interactions between genetic factors play an integral role during awn development and have a wide variety of pleiotropic phenotypic effects. The ampersand symbols stand for interactions between/among genetic factors.

**Table 1 genes-10-00573-t001:** Genes that underlie awn development in major grass species.

Species	Locus Name	Locus Symbol	Associated Makers	Physical Position ^a^	Chr. ^b^	Phenotypic Function	Reference
*Oryza sativa*	*AWN1*	*AN1*	*M6298*, *RM6285*	5.88 Mb (region)	4	Awn formation	Luo et al. [12]
*O. sativa*	*AWN2*	*AN2*	*FM5*, *FM6*	56 kb (region)	4	Awn elongation	Gu et al. [16]
*O. sativa*	*AWN3*	*AN3*	-	-	3	Awn elongation	Takamure et al. [54]
*O. sativa*	*AWN4*	*AN4*	*RT1-8a*T65* *RT1-8c*T65* *RT8-10*T65*	- - -	8	Awn elongation, inflorescence development	Sato et al. [55]
*O. sativa*	*AWN5* (*An5*)	*AN5*	*RZ740* *RG122*	21.2 cM 24.5 cM	4	Awn elongation	Xiong et al. [56]
*O. sativa*	*An-10* [*An5(t)*]	*An10*	*C1677* *G1316*	2.4 cM 2.2 cM	3	Awn elongation	Kubo et al. [57]
*Oryza glumaepatula*	*An-11*(*An7*)	*An11*	*RM3419* *RM289*	3.6 cM 3.6 cM	5	Awn elongation	Matsushita et al. [13]
*O. glumaepatula*	*An-12*(*An8*)	*An12*	*RM261* *RM1359*	12.2 cM 7.1 cM	4	Awn elongation	Matsushita et al. [13]
*Oryzameridionalis*	*An-13*(*An6*)	*An13*	*RM3496*	11.7 cM	8	Awn elongation	Matsushita et al. [14]
*O. meridionalis*	*An-14*(*An9*)	*An14*	*RM8111* *RM8051*	3.0 cM 3.6 cM	1	Awn elongation	Matsushita et al. [14]
*O. meridionalis*	*An-15*(*An10*)	*An15*	*RM237* *RM265*	15.1 cM 3.0 cM	1	Awn elongation	Matsushita et al. [14]
*O. sativa*	*Chromogen for Anthocyanin*	*C*	*XNpb165-1* *XNpb200*	16.4 cM 17.1 cM	6	Lemma, palea, and awn color	Kishimoto et al. [58]
*O. sativa*	*Long and Barbed Awn 1*	*LABA1* (*RAE1*)	*M3*, *RM17242*	34.6 kb (region)	4	Awn elongation	Hua et al. [15]
*O. sativa*	*Grain Number, Grain Length, and Awn Development 1*	*GAD1* (*RAE2*)	*MX14*, *MX16*	6 kb (region)	8	Awn elongation, grain number	Jin et al. [17]
*O. glabberima*	*Regulator of Awn Elongation 3*	*RAE3*	*6KG28331*, *6KG30196*	1.9 Mb (region)	6	Awn elongation	Furuta et al. [6]
*Triticum aestivum*	*Hooded*	*Hd*	*WABM229716*, *WABM117400*, *WABM233735*17.1	0.9 cM	4AS	Awn suppression	Yoshioka et al. [19]
*T. aestivum*	*Tipped 1*	*B1*	*Xgwm291*	1.3 cM	5AL	Awn suppression	Yoshioka et al. [19]
*T. aestivum*	*Tipped 2*	*B2*	*WABM232658* *WABM243094*	1.3 cM 1.9 cM	6BL	Awn suppression	Yoshioka et al. [19]
*Aegilops tauschii*	*Anathera*	*Antr*	*S57615-1* *Xctg211719*	1.3 cM 3.9 cM	5DS	Awn suppression	Nishijima et al. [18]
*Hordeum vulgare*	*Awnless 1*	*Lks1*	*SNP 1_0619* *SNP 1_1533*	133.59 cM 141.56 cM	2HL	Awn suppression	Franckowiak and Lundqvist [59]
*H. vulgare*	*Long Awn2*	*Lks2*	*k06123* *k04151*	0.27 cM 1.0 cM	7HL	Awn elongation	Yuo et al. [1]
*H. vulgare*	*Hooded lemma 1*	*Kap1* (*Knox3*)	*glf3*	25.1 cM	4HS	Hood formation, awn suppression	Müller et al. [60]
*H. vulgare*	*Six-rowed spike 1*	*vrs1*	*e40m36-1110S* *BC12348*	0.01 cM 0.06 cM	2H	Reduces awn length	Komatsuda et al. [61]
*H. vulgare*	*K Suppressor* loci *suKB*	*suKB*	*E34M46* *E35M39*	16.9 cM 9.9 cM	7H	Reduces awn length, Hood formation	Roig et al. [62]
*H. vulgare*	*K Suppressor* loci *suKC*	*suKC*	*E35M44* *E41M46*	0.0 cM 7.8 cM	7H	Reduces awn length, Hood formation	Roig et al. [62]
*H. vulgare*	*K Suppressor* loci *suKD*	*suKD*	*E37M38* *E41M36*	15.9 cM 1.1 cM	5H	Reduces awn length, Hood formation	Roig et al. [62]
*H. vulgare*	*K Suppressor* loci *suKE*	*suKE*	*E40M43* *E36M44*	5.6 cM 0.0 cM	7H	Hood formation	Roig et al. [62]
*H. vulgare*	*K Suppressor* loci *suKF*	*suKF*	*E35M40* *E43M32*	5.3 cM 22.4 cM	7H	Hood formation	Roig et al. [62]

^a^ Physical position: bp, base pair; cM, centimogan; kb, kilobase; Mb, megabase. ^b^ Chr, chromosome.

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
