# Peer review of "Morphological and Genetic Mechanisms Underlying Awn Development in Monocotyledonous Grasses"

_genes, 2019, doi:10.3390/genes10080573_

Round 1

Reviewer 1 Report

The objectives of the manuscript are to review the phenotypic evolution of awns by illustrating morphological and histological characteristics of the awn and physiological changes during awn development, and to assess the genetic basis associated with awn development and variation in major cereal crops.

The title adequately describe the subject of the manuscript and the abstract briefly tell what was done and  summarize the main results and conclusions. The subject is developed logically and effectively and the manuscript is well organized and concise. The conclusions are adequate and supported by the data.  The information is presented in a relatively simple and straightforward manner to be understood by a competent scientist or reader. The manuscript is a useful contribution to knowledge about subjects within the scope of the Journal “Gene”.

Minor revision:

- line 99: Review the incomplete sentence “characteristic features of the grass inflorescence.”

- line 109: “In wheat, there is no rule for awns to be long or short, and these categories are estimated via

 observations [29];”: replace the semicolon with the dot.

- line 365: “on chromosome arm 5AL, and B2 on chromosome arm 6BC [16].” Review the chromosome arm 6BC

- Completely review the format of the References according to the rules of the Journal. The scientific names in Latin should be written in italics.

Some examples: 

- line 450: “SÁNCHEZ-KEN, J.G. Lachnagrostis filiformis (Poaceae: Pooideae: Poeae: Agrostidinae) in Mexico: known  distribution and suppression of lemma awn development in terminal spikelets. Phytotaxa 2018, 350, 223-452 234.” The name of the author should have only the initial capital letters,  and  the Latin name Lachnagrostis filiformis should be in italics.  

- line 484: Triticum aestivum should be in italics  

- lines 488-491:  42. Dong, R.; Dong, D.; Luo, D.; Zhou, Q.; Chai, X.; Zhang, J.; Xie, W.; Liu, W.; Dong, Y.; Wang, Y., et al. Transcriptome Analyses Reveal Candidate Pod Shattering-Associated Genes Involved in the Pod Ventral  Sutures of Common Vetch (Vicia sativa L.). Frontiers in Plant Science 2017, 8, 649, doi:10.3389/fpls.2017.00649.

The title of the paper should be in small letters. 

- line 490:  Sutures of Common Vetch (Vicia sativa L.). Vicia sativa should be in italics 

- line 495: Journal of genetics 2016, 95, 639-646. “genetic” with the initial capital letter 

- line 506: Oryza sativa/O. rufipogon  in italics 

- line 509: Aegilops tauschii in italics 

and so on.

- Table 1: the table should be well formatted

- Figure 3: complete the caption “of genetic factors that underlie awn development in major cereal crops.”:

Author Response

 Dear Reviewer,

We thank you for the time and efforts you have put into assessing the previous version of our manuscript.

Title: Morphological and Genetic Mechanisms Underlying Awn Development in

     Monocotyledonous Grasses

Authors: Fabrice Ntakirutimana and Wengang Xie*

Your comments and suggestions are all valuable and very helpful for revising and improving our manuscript, as well as the important guiding significance to our researches. After studying the comments and suggestions, we have carefully revised our manuscript. Now, we are pleased to submit the revised version to you. In the response, all the comments and suggestions have been answered in a point-by-point fashion. The major changes we have made to the original manuscript are highlighted in colored text.

Thank you again for your attention to our manuscript!

Sincerely yours,

Dr. Wengang Xie 

Point-by-point responses to the comments

Comments and suggestions to the Author: The objectives of the manuscript are to review the phenotypic evolution of awns by illustrating morphological and histological characteristics of the awn and physiological changes during awn development, and to assess the genetic basis associated with awn development and variation in major cereal crops.

The title adequately describes the subject of the manuscript and the abstract briefly tell what was done and summarize the main results and conclusions. The subject is developed logically and effectively and the manuscript is well organized and concise. The conclusions are adequate and supported by the data.  The information is presented in a relatively simple and straightforward manner to be understood by a competent scientist or reader. The manuscript is a useful contribution to knowledge about subjects within the scope of the Journal “Genes”.
Response: Thank you very much for your attention and kind suggestion! We appreciate so much for the warm and encouraging comments from you! According to your suggestions, we have revised and rewritten our manuscript thoroughly.
Specific Comments:
Comment 1: line 99: Review the incomplete sentence “characteristic features of the grass inflorescence.”
Response: Thanks a lot for your attention and kind suggestion! You are right and the part of this sentence is missing. Now, we have revised and rewritten the sentence. Please check line 107.
Comment 2: line 109: “In wheat, there is no rule for awns to be long or short, and these categories are estimated via observations [29];”: replace the semicolon with the dot.

Response: Thanks a lot for the comment and helpful suggestion! We agree with you that the punctuation mark used in this sentence was not correct and thus we have checked and corrected it according to your suggestion (Please check lines 117-118).
Comment 3: line 365: “on chromosome arm 5AL, and B2 on chromosome arm 6BC [16].” Review the chromosome arm 6BC.
Response: Thanks a lot for the comment! It is our fault that the chromosome was not correct due to typing error. Based on your comment, we checked this chromosome in the respective papers and in recognized databases and we have corrected it accordingly (Please check line 441).
Comment 4: Completely review the format of the References according to the rules of the Journal. The scientific names in Latin should be written in italics. 

Response: Thank you very much for your kind suggestion. In fact, both in-text citation and references needed to be checked and corrected according to the rules of the journal, as you suggested. We have deeply checked every reference and citation and corrected them accordingly.

Comments 5: line 450: “SÁNCHEZ-KEN, J.G. Lachnagrostis filiformis (Poaceae: Pooideae: Poeae: Agrostidinae) in Mexico: known  distribution and suppression of lemma awn development in terminal spikelets. Phytotaxa 2018, 350, 223-452 234.” The name of the author should have only the initial capital letters, and the Latin name Lachnagrostis filiformis should be in italics.  .
Response: Thank you very much for your attention and kind suggestion! We agree that this reference was not written according to the journal citation style. Thus, we have corrected this reference basing on your suggestion .
Comment 6: line 484: Triticum aestivum should be in italics  

Response: Thank you very much for your attention and kind suggestion! Based on your suggestions, we have corrected this scientific name. 
Comment 7: lines 488-491:  42. Dong, R.; Dong, D.; Luo, D.; Zhou, Q.; Chai, X.; Zhang, J.; Xie, W.; Liu, W.; Dong, Y.; Wang, Y., et al. Transcriptome Analyses Reveal Candidate Pod Shattering-Associated Genes Involved in the Pod Ventral  Sutures of Common Vetch (Vicia sativa L.). Frontiers in Plant Science 2017, 8, 649, doi:10.3389/fpls.2017.00649.

The title of the paper should be in small letters
Response: Thank you very much for your attention and kind suggestion! Based on your comments and suggestions, we have corrected this reference according to the journal citation style.

Comment 8: line 490:  Sutures of Common Vetch (Vicia sativa L.). Vicia sativa should be in italics 

Response: Thank you very much for your attention and kind suggestion! We have corrected this reference according to the journal citation style .

Comment 9: line 495: Journal of genetics 2016, 95, 639-646. “genetic” with the initial capital letter  

Response: We really thank you for your attention and kind suggestion! We have corrected this reference according to the journal citation style (Please check line 563-543).
Comment 10: line 506: Oryza sativa/O. rufipogon  in italics 
Response: Thank you very much for your attention and kind suggestion! We have corrected this reference according to the journal citation style .
Comment 11:  line 509: Aegilops tauschii in italics 

and so on.
Response: Thank you very much for your attention and kind suggestion! We have corrected this reference according to the journal citation style. In addition, we have checked the whole manuscript and corrected in-text citations and references.

Comment 12: Table 1: the table should be well formatted.

Response: Thank you very much for your attention and kind suggestion! We agree that the table 1 was not in good format. Thus, we have corrected the table 1 according to the journal guidelines

Comment 13: Figure 3: complete the caption “of genetic factors that underlie awn development in major cereal crops.”
Response: Thank you very much for your attention and kind suggestion! We agree that the caption of the Figure 3 was incomplete. Thus, we have completed this caption.

Thank you again for your attention to our manuscript!

Reviewer 2 Report

This work reviews literature concerning morphological and genetic mechanism that influence awn development in grasses. The paper is well written and organized, and it's easy to understand. Its main value is that this subject is rarely undertaken by itself in the literature and the authors made an extensive search about the topic.

However, the general interest of the selected topic is only medium at best. The paper is pretty descriptive and the authors don't emphasize on the importance of the awns for example in grass forage crops, or the quantitative importance for yield under limiting conditions.

There are extensive parts of the paper that do not really contain information of any kind:

- Line 141. If Arabidopsis thaliana does not have awns it is very strange to mention it in this paper.

- Section 4.1 (from line 233 to 277): This section does not include sufficient explanations of methodologies for those who don't know them, nor specific information for those who know the methods. It seems to me as superfluous text.

- Line 287 to 289 (this variability has prompted... to underlying awn development in rice). Those are again a lot of words without information.

- Lines 322-324: Again a lot of wording that do not add significance to the text.

- The conclusion and future perspectives is also quite a deception as it does not contain much information and only general sentences without concretion

Author Response

Dear Reviewer,

We thank you for the time and efforts you have put into assessing the previous version of our manuscript.

Title: Morphological and Genetic Mechanisms Underlying Awn Development in

     Monocotyledonous Grasses

Authors: Fabrice Ntakirutimana and Wengang Xie*

Your comments and suggestions are all valuable and very helpful for revising and improving our manuscript, as well as the important guiding significance to our researches. After studying the comments and suggestions, we have carefully revised our manuscript. Now, we are pleased to submit the revised version to you. In the response, all the comments and suggestions have been answered in a point-by-point fashion. The major changes we have made to the original manuscript are highlighted in colored text.

Thank you again for your attention to our manuscript!

Sincerely yours,

Dr. Wengang Xie

Point-by-point responses to the comments

Comments and suggestions to the Author: This work reviews literature concerning morphological and genetic mechanism that influence awn development in grasses. The paper is well written and organized, and it's easy to understand. Its main value is that this subject is rarely undertaken by itself in the literature and the authors made an extensive search about the topic.

Response: Thank you very much for your attention and kind suggestion! We appreciate so much for the warm and encouraging comments from you! According to your suggestions, we have revised and rewritten our manuscript thoroughly.

However, the general interest of the selected topic is only medium at best. The paper is pretty descriptive and the authors don't emphasize on the importance of the awns for example in grass forage crops, or the quantitative importance for yield under limiting conditions. 

Response: Thank you very much for your attention and sincere comment! According to your comments, we have added information on the importance of awns in forage grasses (Please check lines 42-43). In addition, we have provided information on the role of awns under unfavorable conditions, as you suggested (Please check lines 39-40).

Specific Comments:
Comment 1: Line 141. If Arabidopsis thaliana does not have awns it is very strange to mention it in this paper
Response: Thanks a lot for your attention and sincere comment! You are right Arabidopsis thaliana is an eudicot plant and has no awns. In fact, we used this species to compare flower development in monocots and eudicots to provide an overview on how this process is more complex in grasses (monocots). As no awned grass species is known to exhibit simple flower development to date, this species remains a model crop and useful for many research purposes, including those comparing developmental processes in grasses and eudicots. However, according to your suggestions, we have revised the mentioned statement and provided more clear information. Please check lines 151-154.
Comment 2: Section 4.1 (from line 233 to 277): This section does not include sufficient explanations of methodologies for those who don't know them, nor specific information for those who know the methods. It seems to me as superfluous text.

Response: Thanks a lot for the kind comment and sincere concern! We agree with you that this section does not deeply discuss mentioned methodologies. In fact, we wanted to give to the leaders a brief introduction on the methodologies that have been applied so far to unravel genetic basis of awn development. Discussing these methodologies could have provided much information that may mislead the leaders or divert the main objectives of the manuscript. However, according to your comments we briefly describe these methods such as QTL mapping and GWAS, we also provided references for most recent studies on these methodologies and this could help our leaders to access the information without repeating the information already given by previous researchers (Please check lines 253-258, 278-279, 285-288).
Comment 3: Line 287 to 289 (this variability has prompted... to underlying awn development in rice). Those are again a lot of words without information.
Response: Thanks a lot for the comment! We agree that we provided much unnecessary information in this statement, which could mislead the leaders. Thus, based on your comments, we have revised and rewritten this statement to provide more clear and exact information (Please check lines 298-299).
Comment 4: Lines 322-324: Again a lot of wording that do not add significance to the text. 

Response: Thank you very much for your comment. We agree that we provided much unnecessary information in this statement, which could mislead the leaders. Thus, based on your comments, we have revised and rewritten this conclusion and provided more clear and specific information please see line 286-288.

Comments 5: The conclusion and future perspectives is also quite a deception as it does not contain much information and only general sentences without concretion.

Response: Thank you very much for your attention and sincere comment! We agree that the information provided in the conclusions is not deeper. In fact, we have tried to provide concluding remarks at the end of every section in order to provide idea follow and help our leaders to access easily related information, as the review has several different sections (for example: lines 147-149; 235-236; 286-288; 423-428). Also, in the conclusions, we rewritten some content and have focused on the importance of the literature provided as well as major future perspectives. Please see line 456-469.

Thank you again for your attention to our manuscript!

Reviewer 3 Report

The manuscript ‘Morphological and Genetic Mechanisms Underlying Awn Development in Monocotyledonous Grasses’ by Fabrice Ntakirutimana and Wengang, provides an overview of the current knowledge of key factors and events in grasses awn development. In general, it is a very accessible and well-structured review. However, some points should be drawnclearly, and the summarized insights should be improved based on current publications.

1.    The author should summarize the morphological differences of awn among grass species clearly. Such as the awn diversity in wild and cultivated rice species, compared with wheat and barley. The current version looks disordered.

2.    After the evolution of spikelet in grasses, some species, like maize, do not have the awn in flowers/seeds, the author should mention this diversity in grasses and discuss this point in evolutionary process.

3.    In rice, most wild rice species are known as long awned, whereas cultivated 109 cultivars are awnless or remarkably short-awned’. Why did most cultivated rice cultivars show awnless or remarkably short-awned, genetics or environmental factors? Is it good for agriculture? What is the difference between wild and cultivated cultivars in other crops?

4.    Line 80-82, ‘Determinate inflorescence 80 produces a fixed number of florets, as in rice, and barley. Indeterminate inflorescence generates an 81 unfixed number of lateral florets before senescence, as in wheat and rye’, authors might mix the determinate/indeterminate inflorescence and determinate/indeterminate spikelet.  For example, the barley belongs to indeterminate inflorescence and determinate spikelet. Authors need to introduce them carefully.

5.    The rice an1mutant showed a reduced cytokinin level in awn primordia, and AN2 encodes a cytokinin synthesis protein, Lonely Guy Protein 6 (OsLOGL6), which suggests the role of cytokinin in awn development. The authors should discuss this point at cellular and molecular levels.

6.    Lines 358-362, ‘Furthermore, recombinant inbred lines (RIL) from two barley accessions, 358 Azumamugi and Kanto Nakate Gold displayed several awn development QTLs near Uzu1, Dsp and 359 Vrs1 genes that are known to be involved in the regulating spike development and plant statute traits’. The QTLs only near Uzu1, Dsp and Vrs1 genes, but vrs1mutants did not show the phenotype of awn development. Thus, how authors drew the conclusion that ‘interactions among these genetic factors may be involved in regulating awn development’?

Author Response

Dear Reviewer,

We thank you for the time and efforts you have put into assessing the previous version of our manuscript.

Title: Morphological and Genetic Mechanisms Underlying Awn Development in

     Monocotyledonous Grasses

Authors: Fabrice Ntakirutimana and Wengang Xie*

Your comments and suggestions are all valuable and very helpful for revising and improving our manuscript, as well as the important guiding significance to our researches. After studying the comments and suggestions, we have carefully revised our manuscript. Now, we are pleased to submit the revised version to you. In the response, all the comments and suggestions have been answered in a point-by-point fashion attached below. The major changes we have made to the original manuscript are highlighted in colored text.

We look forward to hearing from you in due time to respond to any further questions and comments you may have.

Thank you again for your attention to our manuscript!

Sincerely yours,

Dr. Wengang Xie

Point-by-point responses to the comments

Comments and suggestions to the Author: The manuscript ‘Morphological and Genetic Mechanisms Underlying Awn Development in Monocotyledonous Grasses’ by Fabrice Ntakirutimana and Wengang, provides an overview of the current knowledge of key factors and events in grasses awn development. In general, it is a very accessible and well-structured review. However, some points should be drawn clearly, and the summarized insights should be improved based on current publications.

Response: Thank you very much for your attention and kind suggestion! We appreciate so much for the warm and encouraging comments from you! According to your suggestions, we have revised and rewritten our manuscript thoroughly.
Specific Comments:
Comment 1:  The author should summarize the morphological differences of awn among grass species clearly. Such as the awn diversity in wild and cultivated rice species, compared with wheat and barley. The current version looks disordered

Response: Thanks a lot for your attention and kind suggestion! You are right and in the previous version looked more disordered. Now, we have revised and rewritten the paragraph, with more clear and easy to understand information. Please check lines 107-119.
Comment 2: After the evolution of spikelet in grasses, some species, like maize, do not have the awn in flowers/seeds, the author should mention this diversity in grasses and discuss this point in evolutionary process.

Response: Thanks a lot for comment and sincere suggestion! We agree with you that a deeper discussion on the evolutionary process of awn diversity in grasses was lacking in the previous version.  Thus, although awns have generally lagged behind in recent studies and thus information is still limited, we have tried to provide some information, according to your suggestions (Please check lines 108-110; and 113-115).
Comment 3: ‘In rice, most wild rice species are known as long awned, whereas cultivated 109 cultivars are awnless or remarkably short-awned’. Why did most cultivated rice cultivars show awnless or remarkably short-awned, genetics or environmental factors? Is it good for agriculture? What is the difference between wild and cultivated cultivars in other crops?.
Response: Thanks a lot for the kind comment and important questions! Based on your comments, we have provided information these questions this subject (Please check lines 44-52).
Comment 4:  Line 80-82, ‘Determinate inflorescence produces a fixed number of florets, as in rice, and barley. Indeterminate inflorescence generates an unfixed number of lateral florets before senescence, as in wheat and rye’, authors might mix the determinate/indeterminate inflorescence and determinate/indeterminate spikelet.  For example, the barley belongs to indeterminate inflorescence and determinate spikelet. Authors need to introduce them carefully. 

Response: Thank you very much for comment and suggestions! We agree that in the provided statements we mixed floral determinacy in respect to the grass inflorescence and spiklets. Thus, we have revisited several previous studies (for instance, Bonnett, 1966) and carefully corrected the previous version, basing on your suggestions (Please check lines 86-89).
Comment 5: The rice an1mutant showed a reduced cytokinin level in awn primordia, and AN2 encodes a cytokinin synthesis protein, Lonely Guy Protein 6 (OsLOGL6), which suggests the role of cytokinin in awn development. The authors should discuss this point at cellular and molecular levels. 

Response: Thank you very much for comment and suggestions! We agree that this information is important for the general content of the manuscript as it covers the main objectives of the study. Thus, we have revisited several previous studies and provided cellular and molecular information on these genes, basing on your suggestions (Please check lines 359-379).

Comment 6: Lines 358-362, ‘Furthermore, recombinant inbred lines (RIL) from two barley accessions, 358 Azumamugi and Kanto Nakate Gold displayed several awn development QTLs near Uzu1, Dsp and 359 Vrs1 genes that are known to be involved in the regulating spike development and plant statute traits’. The QTLs only near Uzu1, Dsp and Vrs1 genes, but vrs1mutants did not show the phenotype of awn development. Thus, how authors drew the conclusion that ‘interactions among these genetic factors may be involved in regulating awn development’?

Response: Thank you very much for your attention and sincere comment! We agree that this information is misleading and, especially, contradict the previous findings. Based on your comments and suggestions, we have revisited several previously published papers and provided clearer information. In addition, we have used these previously published data to add information and provide future perspectives (Please check lines 424-437).

Thank you again for your attention to our manuscript!

Round 2

Reviewer 2 Report

The paper has been substantially improved. I consider it suitable for publication in its present form.